# Pesticide Exposure and Mucocutaneous Symptoms Among Thai Agricultural Workers: A Cross-Sectional Study

**DOI:** 10.3390/ijerph23010097

**Published:** 2026-01-10

**Authors:** Warin Intana, Chime Eden, Weeratian Tawanwongsri

**Affiliations:** 1School of Agricultural Technology and Food Industry, Walailak University, Nakhon Si Thammarat 80160, Thailand; iwarin@wu.ac.th; 2Division of Dermatology, Jigme Dorji Wangchuck National Referral Hospital (JDWNRH), Thimphu 11001, Bhutan; ceden@jdwnrh.gov.bt; 3Division of Dermatology, Department of Internal Medicine, School of Medicine, Walailak University, Nakhon Si Thammarat 80160, Thailand

**Keywords:** pesticide exposure, agricultural workers, occupational dermatoses, mucocutaneous symptoms, personal protective equipment, dermatology life quality index, Thailand

## Abstract

**Highlights:**

**Public health relevance**
This study quantifies the prevalence and clinical patterns of pesticide-attributed mucocutaneous symptoms among agricultural workers in southern Thailand.It evaluates dermatology-specific quality of life using the Dermatology Life Quality Index (DLQI) to characterize the occupational burden of these symptoms.

**Public health significance**
Pesticide-attributed symptoms were reported by 14.6% for skin involvement, 5.3% for ocular involvement, and 0.4% for oral/nasal involvement; symptoms were predominantly mild and intermittent.DLQI scores were generally low overall but were significantly higher among participants reporting skin and ocular symptoms.

**Public health implications**
The findings support strengthening personal protective equipment (PPE) training, practical risk communication, and routine screening for mucocutaneous symptoms within occupational health services.These data can inform surveillance and prevention strategies aimed at reducing pesticide-related mucocutaneous morbidity in Thailand and comparable agricultural settings.

**Abstract:**

Exposure to plant protection products (pesticides) is common among agricultural workers and may represent an underrecognized cause of mucocutaneous disease. We conducted a descriptive cross-sectional survey in agricultural communities in southern Thailand (August–November 2025) to estimate the prevalence, clinical characteristics, and dermatology-specific quality-of-life impact of pesticide-attributed symptoms. Agricultural workers with pesticide use or exposure within the preceding 12 months were recruited via convenience sampling; participants provided consent and completed standardized interviewer-administered questionnaires assessing demographics, pesticide exposure history and application practices, personal protective equipment (PPE) use, self-reported cutaneous and mucosal symptoms (ocular and oral/nasal), and the Dermatology Life Quality Index (DLQI). Of the 354 eligible individuals, 228 participated in the study, and 226 were included in the analyses. The median age was 54 years (interquartile range [IQR], 15), and 82.7% were male. Overall, 14.6% reported pesticide-attributed cutaneous symptoms, 5.3% reported ocular mucosal symptoms, and 0.4% reported oral/nasal mucosal symptoms. Cutaneous manifestations were predominantly symptoms occurring after exposure, with pruritic, erythematous eruptions affecting the arms and hands that typically resolved within 1–7 days after cessation of exposure. Among symptomatic participants, the median DLQI was 0.5 (IQR 3.0); however, DLQI scores were significantly higher among participants who reported pesticide-attributed cutaneous symptoms (*p* < 0.001) and ocular symptoms (*p* < 0.001). These findings suggest that pesticide-associated mucocutaneous effects are generally mild yet clinically meaningful, underscoring the need to strengthen PPE training, risk communication, and occupational health surveillance in agricultural settings.

## 1. Introduction

Global plant protection products (pesticides) use has increased by approximately 46%, with Asia accounting for more than half of global consumption. By 2018, an estimated 4 million tons were applied annually worldwide, comprising predominantly herbicides (56%), fungicides (25%), and insecticides (19%) [1]. In regulatory terms, pesticides used to protect crops and plant products are often referred to as plant protection products, whereas biocides are designed to kill or inhibit harmful organisms but are intended for non-agricultural protective uses (e.g., antifouling coatings and wood preservatives) [2]. In this study, we use the term “pesticides” to denote agricultural plant protection products, which constitute a major subset of biocidal chemicals relevant to farm work exposures. Farmers and other agricultural workers represent a high-risk population because occupational exposure can occur repeatedly and via multiple routes, resulting in both acute and chronic toxicity. Acute pesticide-related effects may include nausea, dizziness/vertigo, and irritant skin reactions, whereas long-term exposure is associated with a broader spectrum of disorders affecting neurologic, respiratory, hepatic, and cutaneous systems [3,4,5]. Pesticides exert toxicity through dermal contact and inhalational exposure, contributing to systemic effects such as neurotoxicity, hepatotoxicity, severe respiratory compromise, and mucocutaneous injury [6,7]. Cutaneous manifestations are often among the earliest and most frequently reported outcomes; these include irritant and allergic contact dermatitis, pigmentary alterations, chloracne, urticaria, and skin malignancies. Selected agents and classes—including glyphosate, paraquat, thiurams, carbamates, and arsenicals—have been implicated through mechanisms involving oxidative stress, direct cytotoxicity, and immune-mediated sensitization pathways. In agricultural settings, inadequate personal protective equipment, high ambient temperatures, and suboptimal work practices may further increase dermal absorption, thereby amplifying the burden of pesticide-related dermatoses [3].

The burden of mucocutaneous conditions among agricultural workers exposed to pesticides has been documented in several studies across Asia, underscoring the ongoing significance of this occupational health issue. In Pakistan, Khan and Damalas (2015) reported that approximately one-third of cotton farmers experienced pesticide-related skin symptoms (33.6%) and ocular irritation, which were primarily attributed to limited use of personal protective equipment (PPE) and unsafe handling practices; for example, 76.4% reported treating spills inadequately, 67.9% used defective sprayers, and 46.5% worked in windy conditions [8]. In Vietnam, Huyen et al. (2020) found that 22.5–29.6% of farmers reported skin symptoms, and 21.3–26.8% reported eye irritation following direct contact with chemical or biological pesticides, alongside suboptimal post-application hygiene practices; notably, only 13.5% reported using eye drops after exposure [9]. In Taiwan, Wu et al. (2024) observed that 26.1% of field crop operators reported allergic contact dermatitis, 5.8% irritant dermatitis, and 12.4% other dermatitis-related conditions [10]. Importantly, drone-based application was associated with an approximately 50% reduction in the likelihood of dermatitis and keratitis compared with ground-based spraying. In a survey of more than 4000 farmers in Thailand, Sapbamrer et al. (2024) reported that 14.8% of the farmers experienced skin symptoms, and 41.2% reported ocular irritation [11]. Dermatitis was positively associated with fungicide exposure (adjusted odds ratio [aOR], 2.03) and molluscicide exposure (aOR, 1.73). The survey further indicated that limited knowledge regarding pesticides and PPE, together with poor adherence to PPE use, was associated with a higher prevalence of reported symptoms among participants. These studies suggest that, despite advances in application technology and regulatory oversight, pesticide-associated mucocutaneous conditions remain prevalent among agricultural workers across Asia, likely driven by sustained chemical exposure, unsafe handling and application practices, and inadequate protective measures.

The Asian region, particularly Thailand, provides an important context for understanding the effects of pesticide exposure on dermatologic and mucosal health. This is attributable to the high prevalence of agricultural work and a hot, humid climate that may enhance dermal absorption and compromise the tolerability of PPE [12]. A substantial proportion of the region is devoted to agriculture; consequently, many individuals are likely to experience occupational pesticide exposure. For example, Sapbamrer et al. (2024) reported cutaneous symptoms in 14.8% of Thai farmers, most commonly erythematous or hypopigmented rashes (14.4%), followed by pustules (1.4%) and ulcers (0.9%) [11]. Agricultural factors associated with skin symptoms included self-production of crops; mixing and/or spraying chemicals; harvest timing; use of fungicides, rodenticides, and molluscicides; backpack spraying; and lower perceived safety and protective behaviors. However, this and similar studies have not consistently delineated specific clinical diagnoses (e.g., irritant vs. allergic contact dermatitis) or characterized other pesticide-related eruptions, such as chemical burns or urticaria, as described in the literature [13,14]. In addition, detailed reporting of clinical severity and dermatology-related quality of life among exposed agricultural workers remains limited. Accordingly, this study aimed to estimate the prevalence of pesticide-associated mucocutaneous symptoms among Thai agricultural workers, assess symptom severity, characterize clinical manifestations, evaluate dermatology-related quality of life, and describe PPE practices during pesticide application. These data may inform targeted occupational health interventions and surveillance strategies to mitigate dermatologic risks among agricultural workers in Thailand.

## 2. Materials and Methods

### 2.1. Study Design and Setting

The Walailak University Ethics Committee reviewed and approved this study (WUEC-25-251-01), which was also prospectively registered with the Thai Clinical Trials Registry (TCTR20250814002). This descriptive cross-sectional study was conducted to evaluate the prevalence, characteristics, and severity of cutaneous adverse effects among agricultural workers exposed to pesticides. The study was carried out in agricultural communities in Nakhon Si Thammarat and Phatthalung provinces in southern Thailand. These provinces were selected because of their high levels of agricultural activity and established collaboration with Walailak University’s Biocontrol Center network, which facilitated organized access to farming communities. Data collection took place between August and November 2025.

### 2.2. Participants

Eligible participants were agricultural workers aged 18–70 years who resided in Nakhon Si Thammarat or Phatthalung provinces, were currently engaged in agricultural work, and had used or been exposed to pesticides within the preceding 12 months. All participants provided informed consent. Participants were excluded if they had a diagnosis of skin conditions unrelated to pesticide exposure (e.g., psoriasis, bullous disorders, vasculitis), were unable to provide valid information because of cognitive impairment or a psychiatric condition, or were unwilling to participate. Pesticide operators/applicators were defined as participants who reported personally mixing/loading and applying pesticides. Recruitment was conducted using convenience sampling through a network established by the Walailak University Biocontrol Center, with assistance from local community leaders.

### 2.3. Questionnaire Development and Validity

We developed a structured questionnaire and trained interviewers to collect information on participants’ age, sex, pesticide exposure, dermatologic symptoms, PPE use, and health-related quality of life. The questionnaire was developed based on a review of the literature and input from two dermatology experts to ensure comprehensive coverage of the study aims. It comprised four main sections: (1) demographic and employment information; (2) pesticide exposure history and characteristics; (3) dermatologic symptoms and severity; and (4) the validated Dermatology Life Quality Index (DLQI), with higher scores indicating greater impairment [15].

Content validity was assessed by two independent dermatology specialists. The Index of Item-Objective Congruence (IOC) was calculated for each item, and items were retained if the IOC was ≥0.5 [16]. A pilot study evaluated reliability using a convenience sample of 30 agricultural workers. Internal consistency was assessed using Cronbach’s alpha and demonstrated acceptable reliability (α = 0.75) [17].

### 2.4. Data Collection and Measurement

Data were collected through face-to-face interviews conducted in a single session with residents of agricultural communities. Five trained research assistants conducted interviews in Thai using a standardized questionnaire and a uniform interview protocol to ensure neutrality and reduce interviewer bias. Eligibility was assessed at the start of the consent process, and participants who met the inclusion criteria provided written informed consent. Each interview lasted approximately 30 min.

Trained research assistants administered a standardized questionnaire at all survey sites and followed a uniform protocol to maintain neutrality and minimize interviewer influence. Information on pesticide exposure was collected using an interviewer-administered questionnaire. Participants were asked to list the kinds of pesticides they used on their farms, which were divided into insecticides, herbicides, fungicides, rodenticides, and molluscicides. Participants also reported their main cultivated crop(s) and the approximate distance from their residence to the nearest pesticide-treated field to characterize potential occupational and residential proximity to pesticide application. Data on specific active ingredients or formulations were not collected, as pesticide use varied across crops and seasons and was primarily reported by product category.

Dermatologic outcomes included participant-reported symptoms (type, frequency, duration, and anatomical site), assessed retrospectively, and dermatology-specific health-related quality of life measured using the DLQI. The DLQI is a 10-item instrument with total scores ranging from 0 to 30, with higher scores indicating greater impairment. Using standard banding, DLQI scores of 0–1, 2–5, 6–10, 11–20, and 21–30 indicate no effect, a small effect, a moderate effect, a very large effect, and an extremely large effect on dermatology-related quality of life, respectively.

### 2.5. Sample Size Calculation

The sample size was calculated using the single-population proportion estimation formula [18]. The calculation was based on the previously reported prevalence of dermatologic symptoms among Thai agricultural workers (14.8%) [11]. The required sample size was 228, assuming a 5% margin of error, a 95% confidence level, and allowing for 15% missing data.

### 2.6. Statistical Analysis

Descriptive statistics were used to summarize study variables. Symptom prevalence was summarized for all respondents. Detailed pesticide-use characteristics and preventive practices were described for operators/applicators. Categorical variables were presented as frequencies and percentages. Continuous variables were presented as mean ± standard deviation or median (interquartile range), as appropriate, based on data distribution. Between-group comparisons of categorical outcomes (e.g., prevalence of pesticide-attributed skin, ocular, and oral/nasal symptoms by sex or pesticide class) were performed using the chi-square test or Fisher’s exact test, as appropriate. Comparisons of continuous outcomes between two independent groups (e.g., age or DLQI by symptom status or pesticide class) were performed using the Wilcoxon rank-sum test (Mann–Whitney U test). Correlations between continuous or ordinal variables (e.g., DLQI and age) were assessed using Spearman’s rank correlation coefficient (ρ). To evaluate associations between pesticide class and the presence of pesticide-attributed skin symptoms among pesticide applicators, logistic regression models adjusted for age and sex were fitted; results are reported as odds ratios (ORs) with 95% confidence intervals (CIs). Statistical significance was defined as a two-sided *p*-value < 0.05. All analyses were performed using SPSS (version 18; SPSS Inc., Chicago, IL, USA) and R (version 4.3.2; R Foundation for Statistical Computing, Vienna, Austria).

## 3. Results

### 3.1. Participant Characteristics and Symptoms

We initially invited 354 individuals to participate; 228 completed the survey (response rate, 64.4%). A total of 2 participants were excluded because of missing data, leaving 226 participants for the final analysis. The median age was 54.0 years (IQR 15.0), and most participants were men (82.7%). The median duration of agricultural work experience was 20.0 years (IQR 15.0). Regarding primary crops, 52.7% reported oil palm, and 38.9% reported durian. Smaller proportions reported pre-existing ocular (5.3%), nasal (1.8%), or pulmonary/bronchial (1.3%) comorbidities. Most participants reported living more than 100 m from fields treated with pesticides (84.1%). Overall, 23.5% of the participants reported spraying or applying pesticides, 14.6% reported pesticide-attributed skin symptoms, 5.3% reported pesticide-attributed ocular symptoms, and 0.4% reported pesticide-attributed oral or nasal symptoms (Table 1).

### 3.2. Pesticide Application Characteristics and Preventive Practices

Among the 53 participants who reported pesticide application, insecticides were the most commonly used pesticide type (79.2%), whereas herbicides and fungicides/plant disease control agents were used with equal frequency (69.8%). The median frequency of pesticide application was 2.0 days per month (IQR 5.0), and the median duration of application was 2.5 h per day (IQR 1.0), over a median of 5.5 years (IQR 6.0). The most commonly reported application methods were backpack sprayers and machine/tractor sprayers. Most participants reported always wearing protective clothing and equipment during application (81.1%), most commonly chemical protective masks and gloves, long-sleeved clothing, and rubber boots. More than 90% of the participants reported cleaning their protective equipment after each use; however, only 26.4% reported ever receiving training on PPE use (Table 2). Post-application hygiene behaviors reported by participants were good; over 90% reported washing their hands immediately after pesticide use, changing clothes, showering, and washing pesticide-contaminated work clothes separately from other clothing.

### 3.3. Characteristics of Pesticide-Related Mucocutaneous Symptoms

Among participants who reported pesticide-attributed skin symptoms (*n* = 33), the most frequently reported manifestations were pruritus (30/33, 90.9%) and erythema (26/33, 78.8%), most commonly affecting the arms (25/33, 75.8%) and hands (20/33, 60.6%). Symptoms occurred sometimes after pesticide exposure in most participants (24/33, 72.7%), and most episodes lasted 1–7 days (17/33, 51.5%); the majority reported improvement after stopping pesticide exposure (29/33, 87.9%) (Table 3). Among participants with pesticide-attributed ocular symptoms (*n* = 12), burning/stinging (10/12, 83.3%) and itching (9/12, 75.0%) were most common. Symptoms occurred sometimes after pesticide exposure in most participants (10/12, 83.3%) and usually resolved within 7 days (11/12, 91.6%); all participants reported improvement after avoiding exposure (12/12, 100%) (Table 4). Only one participant (0.4%) reported pesticide-attributed oral/nasal symptoms, describing intermittent nasal burning with episodes lasting 1–7 days and improvement after avoidance.

### 3.4. Dermatology-Related Quality of Life Among Symptomatic Participants (DLQI)

Among participants who indicated that they experienced pesticide-related symptoms affecting the skin or mucosal membranes of the eyes or oral/nasal cavities (*n* = 36), the median DLQI total score was 0.5 (IQR, 0–3.0; range, 0–11). Most symptomatic participants reported a DLQI in the “no effect” or “small effect” groups, while only a few reported moderate to very large effects on their dermatology-related quality of life (Table 5).

### 3.5. Associations Between Participant Characteristics, Pesticide Use, and Dermatology-Related Quality of Life

Participant characteristics were evaluated in relation to mucocutaneous symptoms and dermatology-related quality of life. The prevalence of pesticide-attributed skin symptoms was similar in females (6/39, 15.4%) and males (27/187, 14.4%) (*p* = 1.000). Pesticide-attributed ocular symptoms were reported by 7.7% of females (3/39) and 4.8% of males (9/187) (*p* = 0.736), while pesticide-attributed oral/nasal symptoms were rare (female: 1/39, 2.6%; male: 0/187) (*p* = 0.173). Age was not associated with pesticide-attributed skin symptoms (Wilcoxon *p* = 0.341), ocular symptoms (*p* = 0.365), or oral/nasal symptoms (*p* = 0.105). The DLQI did not differ by sex (median 0 in both groups; *p* = 0.778). By contrast, the DLQI was higher among participants who reported pesticide-attributed skin symptoms (*p* < 0.001) and ocular symptoms (*p* < 0.001). The DLQI was not correlated with age (Spearman’s ρ = −0.054, *p* = 0.422).

Among participants who personally applied pesticides (n = 53), symptom prevalence and DLQI scores were compared by pesticide class (herbicides, insecticides, and fungicides). Skin symptoms were reported by 54.1% of herbicide users (20/37) vs. 43.8% of non-users (7/16) (*p* = 0.491), 50.0% of insecticide users (21/42) vs. 54.5% of non-users (6/11) (*p* = 0.788), and 51.4% of fungicide users (19/37) vs. 50.0% of non-users (8/16) (*p* = 0.928). Ocular symptoms were reported by 13.5% of herbicide users (5/37) vs. 31.2% of non-users (5/16) (*p* = 0.148), 21.4% of insecticide users (9/42) vs. 9.1% of non-users (1/11) (*p* = 0.667), and 21.6% of fungicide users (8/37) vs. 12.5% of non-users (2/16) (*p* = 0.704). DLQI scores did not differ by herbicide use (*p* = 0.092), insecticide use (*p* = 0.262), or fungicide use (*p* = 0.933). In age- and sex-adjusted logistic regression models, pesticide class was not associated with skin symptoms (herbicide: OR 1.43, 95% CI 0.41–5.12; insecticide: OR 0.87, 95% CI 0.20–3.65; fungicide: OR 0.95, 95% CI 0.28–3.22).

## 4. Discussion

In this descriptive cross-sectional study of Thai agricultural workers, we identified a mild but clinically meaningful self-reported burden of pesticide-associated mucocutaneous symptoms. Overall, 14.6% of the participants reported pesticide-attributed skin symptoms, 5.3% reported ocular symptoms, and 0.4% reported oral or nasal symptoms. Symptom prevalence was assessed in all respondents (n = 226), with additional descriptive analyses among pesticide applicators (n = 53) and symptomatic participants (n = 36). Most symptomatic participants described symptoms occurring sometimes after exposure, with pruritic, erythematous eruptions on the arms and hands that typically resolved within 1–7 days after cessation of pesticide exposure, and nearly 90% reported improvement after avoiding further exposure. Although DLQI scores were low overall (median, 0.5; IQR, 0–3.0), moderate-to-very large dermatology-related quality-of-life impacts were observed in subgroups, and the DLQI was significantly higher among participants reporting skin and ocular symptoms. By contrast, symptom prevalence and DLQI scores did not differ by sex, and age was not associated with symptoms or the DLQI in this sample. Among applicators, symptom prevalence did not differ significantly by pesticide class (herbicides, insecticides, and fungicides), although these analyses were limited by sample size and non-exclusive use across pesticide categories. Despite frequent self-reported PPE use and post-exposure hygiene practices, mucocutaneous symptoms still occurred, indicating that pesticide-related dermatoses remain an occupational health concern and that current protective practices may be insufficient to fully prevent adverse mucocutaneous outcomes in these settings.

The prevalence and pattern of mucocutaneous symptoms observed in this cohort are broadly comparable with, yet somewhat milder than, those reported in previous studies of agricultural workers in Asia and other regions. The proportion of participants who experienced pesticide-attributed skin symptoms (14.6%) was similar to the 14.8% reported in a nationwide Thai survey; however, that survey predominantly described hyperpigmented and erythematous rashes and reported substantially higher ocular symptom prevalence than that observed in our cohort (41.2% vs. 5.3%) [11]. Higher prevalences of skin and ocular symptoms have also been reported among Pakistani cotton farmers (33.6% reported skin and eye irritation) and Vietnamese farmers (22.5–29.6% reported skin symptoms and 21.3–26.8% reported eye irritation), particularly in settings characterized by limited PPE use and suboptimal post-application hygiene practices [8,9]. Among Taiwanese field crop operators, approximately one-quarter experienced dermatitis, and ground-based pesticide application was associated with a higher likelihood of dermatitis and keratitis than drone-based application [10]. Studies focusing on contact dermatitis frequently indicate a higher burden and more severe clinical phenotypes than those reported here; for instance, studies on cohorts from India and Indonesia identified recurrent pesticide-related dermatitis and a dose–response relationship with spraying frequency and non-use of PPE [19,20]. Similarly, the high prevalence of work-related eczema and urticaria among vocational agriculture students, strongly associated with atopy, suggests that these occupational dermatoses can be readily recognized and reported [21]. By contrast, most symptomatic participants in our study reported intermittent, self-limited eruptions with low DLQI scores, which may reflect lower-intensity exposure, comparatively high PPE use and hygiene practices among applicators, and under-recognition of the true disease burden.

Beyond prevalence estimates, both intrinsic (host susceptibility) and extrinsic (modifiable exposure-related) determinants of pesticide-associated mucocutaneous adverse effects are of particular relevance. Previous studies have reported associations between work-related eczema and urticaria and atopy-susceptible populations, especially individuals with a history of respiratory allergy or eczema. These findings support the notion that an inherently compromised epidermal barrier and Th2-biased immune responses may heighten vulnerability to pesticide-induced dermatoses compared with non-atopic individuals [19,20,21]. With respect to modifiable exposure-related factors, multiple determinants have been linked to occupational dermatoses, including mixing and spraying pesticides—particularly when performed by the same individual—use of backpack sprayers, longer application work, and PPE non-use. Consistent evidence also highlights the importance of PPE use and post-application hygiene to reduce residual pesticide contamination on skin and clothing [3,8,9]. Despite generally favorable reported protective behaviors, the occurrence of symptoms nevertheless suggests that factors beyond PPE and hygiene may contribute. Future studies should evaluate whether mucocutaneous outcomes cluster by specific pesticide products or chemical classes and should account for formulation-related irritants and sensitizers by systematically recording product names, active ingredients, and relevant co-formulants, alongside objective exposure assessment and clinical verification. In our study, most pesticide applicators reported using some form of PPE and regularly cleaning their equipment. However, only approximately one-quarter reported receiving training on PPE use, which is critical for ensuring correct selection, consistent use, and effective decontamination practices, thereby minimizing ongoing dermal and mucosal exposure after application and reducing secondary exposure. Mechanistically, experimental and epidemiologic evidence suggests that pesticides and their formulations can provoke irritant and allergic responses. Pesticide exposure has been associated with alterations in stratum corneum lipids, increased transepidermal water loss, and oxidative stress and can induce proinflammatory cytokine secretion, contributing to irritant or allergic contact dermatitis, urticaria, and—when aerosolized—conjunctival and corneal irritation [7,13,22,23]. Co-formulants such as surfactants, solvents, and emulsifiers are also relevant, as they may be poorly tolerated on mucocutaneous surfaces and can enhance dermal penetration of active ingredients, thereby increasing irritancy and the potential for contact urticaria [4,24]. Thus, these data support a model in which genetically predisposed individuals may be at increased risk of pesticide-associated mucocutaneous disease, particularly under conditions of long-term exposure and incomplete protection.

Effective prevention of pesticide-associated mucocutaneous harm requires multifaceted strategies encompassing personal protective behaviors, risk communication, regulatory measures, and adjunctive protective interventions. Consistent use of appropriate PPE—including gloves, long-sleeved shirts, and long pants—substantially reduces dermal exposure and is associated with lower risks of skin disease and pesticide-associated genotoxicity among agricultural workers; however, uptake remains limited in many low- and middle-income settings because of constrained resources, discomfort, and insufficient risk awareness [4,25,26]. Safe handling practices, including correct mixing procedures, avoidance of direct skin contact with concentrates, and appropriate storage and disposal, are likewise essential and can be strengthened through targeted training and education programs [25]. Effective risk communication—through clear labeling, pictogram-based signage, community engagement, and accessible digital resources—should be tailored to heterogeneous literacy levels to ensure that safety information is understood and translated into practice. Structured awareness programs may further reinforce knowledge regarding exposure risks and the benefits of PPE and post-application hygiene measures [25,27]. At the systems level, robust regulatory oversight is needed to detect early toxicity signals and guide risk mitigation, including mandatory pre- and post-marketing risk assessment, together with ongoing surveillance of exposed workers (e.g., periodic medical examinations and biomonitoring) [22,28]. Complementing these approaches, emerging evidence suggests that certain natural compounds (e.g., oleuropein aglycone from extra virgin olive oil) may exert cytoprotective effects against pesticide-induced oxidative stress in skin cells, and barrier creams or protective lotions may offer an additional layer of protection by supporting stratum corneum integrity; however, their effectiveness in real-world agricultural settings requires further evaluation [4,24]. Thus, optimization of PPE and hygiene, strengthened education and communication, enforceable regulation with integrated surveillance, and exploration of adjunctive protective agents and non-chemical alternatives represent complementary pillars for reducing the mucocutaneous burden of pesticide exposure.

This study has several limitations. First, the cross-sectional design precludes temporal and causal inference between pesticide exposure and mucocutaneous outcomes; longitudinal or repeated surveys are needed to clarify directionality and dose–response relationships. Second, the characteristics of exposure and reported symptoms were derived from self-assessment, potentially resulting in recall or reporting bias. This approach also raises concerns about potential confounding or misattribution of symptoms to other factors, such as insect bites, heat or sweating, friction or occlusion from PPE, irritants or allergens from plants, or other chemical exposures. Accordingly, these alternative explanations cannot be discounted in the current study. Although we used a structured interviewer-administered questionnaire with clear recall periods and symptom descriptors, future studies should incorporate objective exposure assessment (e.g., observation, work logs, and biomonitoring) and standardized clinical evaluation with validated symptom measures. In addition, indirect or secondary exposure pathways among non-applicators (e.g., pesticide drift from nearby fields, re-entry into recently treated areas, or take-home exposure via contaminated clothing or equipment) were not systematically assessed, which may have resulted in exposure misclassification. Third, the sample was drawn from agricultural communities in a single region of southern Thailand, which may limit generalizability to other agroecological zones, crop systems, and higher-intensity exposure settings; multi-site studies including high-risk subgroups are warranted. Fourth, the data were collected during a single period of the year, so seasonal variations in climate, work intensity, and pesticide use were not captured; repeated measurements across seasons would provide a more complete exposure profile. In addition, information on the participants’ educational attainment and detailed employment history was not collected, which may have limited our assessment of socioeconomic or knowledge-related factors influencing pesticide use, protective behaviors, and risk perception. Finally, while the questionnaire was based on previous studies and reviewed by experts, and although it was pre-tested, it has not undergone formal validation in this population. Future research should assess its reliability and validity, including methods such as test–retest and construct validation.

## 5. Conclusions

The findings of this cross-sectional study of Thai agricultural workers indicate that pesticide exposure is associated with a quantifiable burden of self-reported mucocutaneous symptoms, predominantly intermittent pruritic and erythematous lesions affecting the upper extremities, with generally low but occasionally clinically meaningful impairment in dermatology-specific quality of life. While symptom prevalence did not differ by age, sex, or pesticide class in the exploratory analyses, dermatology-related quality of life was significantly worse among participants who reported skin and ocular symptoms, underscoring the clinical relevance of these outcomes. Despite widespread reporting of PPE use and generally favorable post-exposure hygiene practices, pesticide-related skin and ocular symptoms remained prevalent and clinically meaningful. These findings suggest that current personal protection strategies and associated education or training may be insufficient to fully mitigate risk. Accordingly, more systematic practice and programmatic changes may be required, including improving PPE availability and correct use, strengthening practical education and risk communication, and integrating routine assessment of skin and ocular health within occupational health services for agricultural workers. Further multicenter longitudinal studies incorporating objective exposure assessment, standardized clinical evaluation of mucocutaneous signs and symptoms, and fully validated instruments are needed to clarify causal pathways, capture seasonal variability, and support evidence-based regulatory and preventive initiatives tailored to agricultural living and working conditions in Thailand and similar contexts.

## Figures and Tables

**Table 1 ijerph-23-00097-t001:** Participant characteristics (*n* = 226).

Characteristics	Value
Sociodemographic characteristics	
Age (years), median (IQR)	54.0 (15.0)
Age group (years), *n* (%)	
18–39	36 (15.9)
40–59	127 (56.2)
60–70	63 (27.9)
Sex, *n* (%)	
Male	187 (82.7)
Female	39 (17.3)
Ethnicity, *n* (%) Thai	226 (100.0)
Occupational and exposure characteristics	
Years of agricultural work, median (IQR)	20.0 (15.0)
Duration of agricultural work, *n* (%)	
<10 years	53 (23.5)
10–19 years	58 (25.7)
≥20 years	115 (50.8)
Personally sprays/applies pesticides, *n* (%)	53 (23.5)
Distance from pesticide-treated field to home, *n* (%)	
<50 m	17 (7.5)
50–100 m	19 (8.4)
>100 m	190 (84.1)
Agricultural characteristics	
Main crop cultivated, *n* (%)	
Oil palm	119 (52.7)
Durian	88 (38.9)
Mangosteen	8 (3.5)
Other crops ^1^	11 (4.9)
Health history and symptoms, *n* (%)	
Pre-existing ocular comorbidities ^2^	12 (5.3)
Pre-existing nasal cavity–related comorbidities ^3^	4 (1.8)
Pre-existing pulmonary or bronchial comorbidities ^4^	3 (1.3)
Pesticide-attributed skin symptoms	33 (14.6)
Pesticide-attributed ocular symptoms	12 (5.3)
Pesticide-attributed oral or nasal symptoms	1 (0.4)

Note. Percentages were calculated using the total sample (*n* = 226). IQR, interquartile range; m, meter. ^1^ Other crops included rambutan (*n* = 3, 1.3%), rubber (*n* = 4, 1.8%), vegetables (including chili and pumpkin; *n* = 3, 1.3%), and coconut (*n* = 1, 0.4%). ^2^ Overall, 12 participants (5.3%) reported pre-existing ocular comorbidities, most commonly pterygium (*n* = 3, 1.3%), cataract (*n* = 2, 0.9%), and pinguecula (*n* = 2, 0.9%). Single cases of previous cataract surgery, epiphora, glaucoma, hyperopia, and diabetes were reported (each *n* = 1, 0.4%). Non-responses were classified as absence of ocular comorbidities. ^3^ Pre-existing nasal-cavity-related comorbidities were uncommon; only 4 participants (1.8%) reported any nasal disease, whereas 222 (98.2%) reported none. ^4^ All pulmonary or bronchial comorbidities were chronic obstructive pulmonary disease.

**Table 2 ijerph-23-00097-t002:** Characteristics of participants who personally applied pesticides (*n* = 53).

Characteristics	Category	Value
Pesticide type, *n* (%)	Herbicides	37 (69.8)
	Insecticides	42 (79.2)
	Fungicides/plant disease control agents	37 (69.8)
Frequency (days/month), median (IQR)	—	2.0 (5.0)
Duration (hours/day), median (IQR)	—	2.5 (1.0)
Years of pesticide use, median (IQR)	—	5.5 (6.0)
Method of pesticide application, *n* (%)	Backpack sprayer	30 (56.6)
	Machine/tractor sprayer	29 (54.7)
	Hand-mixing	1 (1.9)
PPE use during application, *n* (%)	Always	43 (81.1)
	Sometimes	6 (11.4)
	Never	4 (7.5)
PPE items used, *n* (%)	Chemical protective mask	49 (92.5)
	Rubber gloves	41 (77.4)
	Goggles/eye shield	42 (79.2)
	Long-sleeved protective clothing	44 (83.0)
	Rubber boots	47 (88.7)
	Head cover/hood	32 (60.4)
Condition of PPE/clothing, *n* (%)	Good condition	46 (86.8)
	Damaged	7 (13.2)
Cleaning PPE, *n* (%)	After every use	49 (92.5)
	Sometimes	3 (5.6)
	Never	1 (1.9)
PPE training received, *n* (%)	—	14 (26.4)
Post-application practices, *n* (%)	Wash hands with soap	50 (94.3)
	Change clothes immediately	51 (96.2)
	Shower immediately	51 (96.2)
	Wash work clothes separately	48 (90.6)

Note. Percentages were calculated among operators/applicators (*n* = 53). IQR, interquartile range; *n*, number; PPE, personal protective equipment.

**Table 3 ijerph-23-00097-t003:** Characteristics of abnormal skin symptoms related to pesticide exposure (*n* = 33).

Characteristics	Category	Value
Skin symptom type, *n* (%)	Itching	30 (90.9)
	Erythema	26 (78.8)
	Burning, pain, or stinging	14 (42.4)
	Scaling/peeling	6 (18.2)
	Blistering	2 (6.1)
Anatomical site affected, *n* (%)	Arms	25 (75.8)
	Hands	20 (60.6)
	Legs	14 (42.4)
	Feet	10 (30.3)
	Trunk	8 (24.2)
	Face/neck	3 (9.1)
Frequency after pesticide exposure, *n* (%)	Sometimes	24 (72.7)
	Always	8 (24.2)
	Not sure	1 (3.1)
Duration of symptoms per episode, *n* (%)	<1 day	5 (15.2)
	1–7 days	17 (51.5)
	>7 days	11 (33.3)
Change after stopping exposure, *n* (%)	Improved	29 (87.9)
	Not sure	2 (6.1)
	Unchanged	1 (3.0)
	Worsened	1 (3.0)

**Table 4 ijerph-23-00097-t004:** Characteristics of abnormal eye symptoms related to pesticide exposure (*n* = 12).

Characteristics	Category	Value
Ocular symptom type, *n* (%)	Burning/stinging eyes	10 (83.3)
	Itching eyes	9 (75.0)
	Redness	5 (41.7)
	Excessive tearing	3 (25.0)
	Transient blurred vision	3 (25.0)
	Photophobia	2 (16.7)
Frequency after pesticide exposure, *n* (%)	Sometimes	10 (83.3)
	Always	2 (16.7)
Duration of symptoms per episode, *n* (%)	<1 day	7 (58.3)
	1–7 days	4 (33.3)
	>7 days	1 (8.3)
Direct eye exposure to pesticides, *n* (%)	—	2 (16.7)
Change after avoiding pesticide exposure, *n* (%)	Improved	12 (100.0)

**Table 5 ijerph-23-00097-t005:** Dermatology Life Quality Index (DLQI) among participants with pesticide-related symptoms (*n* = 36).

Measures	Value
DLQI total score, median (IQR)	0.5 (3.0)
DLQI total score, range	0–11
DLQI category, *n* (%)	
0–1 (no effect)	22 (61.1)
2–5 (small effect)	11 (30.6)
6–10 (moderate effect)	2 (5.6)
11–20 (very large effect)	1 (2.8)
21–30 (extremely large effect)	0 (0.0)

Note. DLQI, Dermatology Life Quality Index; IQR, interquartile range.

## Data Availability

All data supporting the findings of this study are included within the manuscript. Additional details or clarifications can be provided upon reasonable request to the corresponding author.

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
