# Peer review of "Pesticide Exposure and Mucocutaneous Symptoms Among Thai Agricultural Workers: A Cross-Sectional Study"

_ijerph, 2026, doi:10.3390/ijerph23010097_

Round 1

Reviewer 1 Report

Comments and Suggestions for Authors

The paper was well written and concerned a relevant and important area of exposure science. I do have a few concerns:

The results and discussion focus on the participants who were occupationally exposed during the mixing/loading and application of pesticides (operators). This is fair, but reduces the sample size to 53. I think the discussion needs to highlight this and it should be clarified whether the reported statistics on adverse effects relate just to the operators or to the whole population of 226 respondents.

No mention of possible confounding factors is made, e.g. could some of the symptoms be attributable to factors other than pesticides, e.g. the regions have high populations of mosquitoes. Can you discount these in the discussion? 

I agree with and applaud your openness in highlighting the weaknesses of the study in the conclusion and support the plans for future work. This will be needed to explore exactly why the adverse effects are occurring when the level of protection and hygiene reported by the operators were actually pretty good. One thing which could be explored is whether the adverse effects are attributable to particular pesticides or products.

Some recommendations for rewording 

You use the term "Biocides." Whilst pesticides for agricultural use could fall under this broad term, in relation to plant protection products, the term biocides usually refers to specific formulation ingredients (preservatives) which protect the formulated product from microbial degradation (interestingly some of these are known skin sensitisers). You also use the term pesticides at various points in the manuscript, I suggest for clarity and consistency, you replace the term biocides with pesticides (or plant protection products) throughout.    

LINE 129: "The study will take place in agricultural 129 communities" suggests the study has not yet been carried out. Change to "took place"?

LINE 202: "14.6% reported that they had ever experienced abnormal symptoms." The word ever is superfluous and removing it improves clarity.

Author Response

Reviewer 1

  1. The paper was well written and concerned a relevant and important area of exposure science. I do have a few concerns: The results and discussion focus on the participants who were occupationally exposed during the mixing/loading and application of pesticides (operators). This is fair, but reduces the sample size to 53. I think the discussion needs to highlight this and it should be clarified whether the reported statistics on adverse effects relate just to the operators or to the whole population of 226 respondents.

Reply

Thank you for this comment. We would like to clarify that the prevalence of self-reported abnormal symptoms (skin: 33/226, 14.6%; eye: 12/226, 5.3%; oral/nasal: 1/226, 0.4%) was calculated using the full study population (n = 226), as presented in Table 1. The subgroup of 53 participants refers specifically to respondents who personally mixed/loaded and/or applied pesticides (operators/applicators), for whom we provide additional detailed information on pesticide-use characteristics (e.g., product types, routes/means of exposure, and preventive practices). To prevent misunderstanding, we revised the Methods, Results, and Discussion to explicitly distinguish (i) analyses based on all respondents (n = 226) from (ii) operator-specific descriptive findings (n = 53), and we added clear denominator statements in the text and table footnotes.

  1. No mention of possible confounding factors is made, e.g. could some of the symptoms be attributable to factors other than pesticides, e.g. the regions have high populations of mosquitoes. Can you discount these in the discussion?

Reply

Thank you for this comment. We agree that the reported symptoms may reflect non-pesticide factors (e.g., insect bites, pre-existing dermatoses, heat/sweating, and other farm/environmental irritants). Given the cross-sectional design and self-reported outcomes, we cannot exclude residual confounding or misattribution. We have revised the Discussion/Limitations to acknowledge these potential confounders, avoid causal language, and highlight the need for future studies with clinical verification and objective exposure assessment.

  1. I agree with and applaud your openness in highlighting the weaknesses of the study in the conclusion and support the plans for future work. This will be needed to explore exactly why the adverse effects are occurring when the level of protection and hygiene reported by the operators were actually pretty good. One thing which could be explored is whether the adverse effects are attributable to particular pesticides or products.

Reply

Thank you for this suggestion. We agree that symptoms occurred despite generally favorable self-reported PPE use and hygiene, warranting evaluation of product- or class-specific contributors. We revised the Discussion to highlight future studies assessing associations with specific pesticide products/classes (including formulation components) using more detailed exposure data and objective assessment where feasible. We also clarified that the current study was not designed or powered for robust product-stratified inference, and we therefore present this as a priority for future research.

  1. Some recommendations for rewording. You use the term "Biocides." Whilst pesticides for agricultural use could fall under this broad term, in relation to plant protection products, the term biocides usually refers to specific formulation ingredients (preservatives) which protect the formulated product from microbial degradation (interestingly some of these are known skin sensitisers). You also use the term pesticides at various points in the manuscript, I suggest for clarity and consistency, you replace the term biocides with pesticides (or plant protection products) throughout.

Reply

We thank the reviewer for this important clarification. We agree that the term “biocides” may be interpreted as referring to disinfectants or preservative-type formulation ingredients rather than agricultural plant protection chemicals. To improve technical accuracy, clarity, and consistency, we have replaced the term “biocides” with “pesticides” throughout the manuscript, including the title, abstract, keywords, main text, tables, and conclusions. At first mention, we now specify “pesticides (plant protection products)” to define the exposure scope relevant to agricultural use.

  1. LINE 129: "The study will take place in agricultural 129 communities" suggests the study has not yet been carried out. Change to "took place"?

Reply

Revised as suggested. This and related tense/grammar issues were corrected during professional English editing.

  1. LINE 202: "14.6% reported that they had ever experienced abnormal symptoms." The word ever is superfluous and removing it improves clarity.

Reply

Revised as suggested; minor wording refinements were also addressed during professional English editing.

Reviewer 2 Report

Comments and Suggestions for Authors

Dear authors,

The study investigated the prevalence work exposure to Biocide and related health effects, notably mucocutaneous symptoms among Thai agricultural workers. It is relevant with important implications in public health management of biocide toxicity among farmers. The authors are encouraged to improve the quality of the manuscript using the guidelines below in addition to some remarks made in the main document. Globally, the manuscript requires major revisions to meet publication standards. The authors are encouraged to improve the quality of the manuscript by addressing the detailed comments below and the specific remarks provided in the main document.

Abstract

  1. Lines 41-45, the summary on the method of the study is poorly done, it should be improved.
  2. Lines 46-47, the authors should reformulate the statement addressing whether ‘5.3% ocular symptoms and 0.4% oral or nasal symptoms’ are integral part of mucocutaneous symptoms.
  3. Many of the key words (e.g. Pesticides; Dermatology Life Quality Index; Thailand) do not tight more with the study. This section should be reviewed accordingly.

Introduction

  1. Lines 64-68: reformulate the sentences to avoid redundancy.
  2. Lines 72 – 75: review and improve the sentence formulation.
  3. Lines 78 – 83: review and improve the sentence formulation.
  4. Lines 87 – 89: review and improve the sentence formulation.
  5. Lines 91 – 97: review and improve the sentence formulation.
  6. Biocide is one the main key words of the study. Its nature, important and health impact should be well captured in the introduction.

Materials and Methods

  1. The authors should consider including the nature of biocide used by the farmers, as this can significantly influence resulting mucocutaneous symptoms experienced by the users.
  2. The statement on ‘two specialists in dermatology’ is redundantly presented, see lines 149 – 150.
  3. The authors should specify which language was used during the interview.
  4. Lines 169 – 171: review and improve the sentence formulation.
  5. The authors calculated a minimum sample size of 228 participants. But obtained only 226 individuals. This therefore limits the statistical power of the study, and questions the conclusions of the findings.
  6. The main crop cultivated, and distance from pesticide-treated field to home should be part of the information presented in the questionnaire.
  7. The section on sample size should be presented earlier and separated from that of Statistical analysis.
  8. The section Statistical analysis is poorly done with sentence without verb. There is a mixture of qualitative and quantitative data expression, and all these were grouped into descriptive analysis. Such an inconsistency should be revisited.

Results and Discussion

  1. Sub-titles should be added to the results section.
  2. The Table 1 should be reorganized with age range, exposure duration levels, educational levels, both sexes, etc. Also, data on participants’ race, education, and employment history should be presented.
  3. Line 225, the word ‘reasonable’ should corrected to ‘good, very good, etc.’
  4. Table 2 -4: present it as a 3 columns table, with the 1st being the variable.
  5. Table 2: the route of pesticide exposure signifies the way through which this chemical enters the human system. I suggest the authors to rather use ‘the mean of pesticide application’.
  6. Lines 230 – 241: quantitative figures should be added when commenting the findings.
  7. Out of the 226 farmers interviewed, only 53 applied biocides. It will be important to find out whether the remaining portion used or were directly or indirectly exposed biocides, if yes how?
  8. The authors should present the findings on intermittent mucocutaneous symptoms pruritic, erythematous  skin 
  9. As part of the findings, the authors should present how independent variable such age, sex, education, etc. influence other factors (mucocutaneous symptoms, DLQI).
  10. The mucocutaneous symptoms are susceptible to be influenced by the type of pesticide used. The authors should present findings with respect to this.

Conclusions

  1. The key findings of the study should be clearly presented prior to recommendations.

General comment

  1. Review the English language, harmonizing the caps lock and abbreviations. The findings should be presented using the past tense.
  2. The section Highlights is not an integral section of the author’s guidelines of IJERPH.

References

  1. This section should be reviewed, harmonized according to IJERPH the author’s guidelines.

Comments on the Quality of English Language

The English Language must be reviewed throughout the manuscript.

Author Response

Reviewer 2

  1. The study investigated the prevalence work exposure to Biocide and related health effects, notably mucocutaneous symptoms among Thai agricultural workers. It is relevant with important implications in public health management of biocide toxicity among farmers. The authors are encouraged to improve the quality of the manuscript using the guidelines below in addition to some remarks made in the main document. Globally, the manuscript requires major revisions to meet publication standards. The authors are encouraged to improve the quality of the manuscript by addressing the detailed comments below and the specific remarks provided in the main document.

Reply

Thank you for recognizing the public health relevance of our study on biocide exposure and mucocutaneous symptoms among Thai agricultural workers. In response, we conducted a major revision and systematically addressed all detailed comments in the main document, improving clarity, the methodological description, results presentation, and the discussion. We believe the revised manuscript now better meets publication standards. The language was also revised for clarity and readability (MDPI English Editing).

2 Abstract

2.1 Lines 41-45, the summary on the method of the study is poorly done, it should be improved.

Reply

The abstract methods were revised to clearly specify the study design, participant eligibility, data collection approach, and key variables assessed, improving clarity and alignment with the Methods section.

2.2 Lines 46-47, the authors should reformulate the statement addressing whether ‘5.3% ocular symptoms and 0.4% oral or nasal symptoms’ are integral part of mucocutaneous symptoms.

Reply

The abstract was revised to explicitly classify ocular and oral/nasal symptoms as mucosal components of the mucocutaneous outcomes, eliminating ambiguity.

2.3 Many of the key words (e.g. Pesticides; Dermatology Life Quality Index; Thailand) do not tight more with the study. This section should be reviewed accordingly.

Reply

Thank you for the comment. We have revised the keywords to better reflect the central focus of the study. The keywords now emphasize pesticide exposure among agricultural workers, mucocutaneous and occupational dermatologic outcomes, PPE use, and dermatology-specific quality of life, improving relevance and indexing accuracy.

  1. Introduction

Lines 64-68: reformulate the sentences to avoid redundancy.

Lines 72 – 75: review and improve the sentence formulation.

Lines 78 – 83: review and improve the sentence formulation.

Lines 87 – 89: review and improve the sentence formulation.

Lines 91 – 97: review and improve the sentence formulation.

Biocide is one the main key words of the study. Its nature, important and health impact should be well captured in the introduction.

Reply

Thank you. We revised the Introduction (Lines 64–97) to reduce redundancy and improve sentence formulation and flow. We also strengthened the background on biocides/pesticides by adding a brief terminology clarification distinguishing biocides from agricultural plant protection products (pesticides) and by emphasizing the relevance of these exposures to mucocutaneous health impacts in agricultural workers. In addition, duplicated statements were removed and key epidemiologic evidence was streamlined. The manuscript was further edited for English language, clarity, and readability (MDPI English Editing).

4 Materials and Methods

4.1 The authors should consider including the nature of biocide used by the farmers, as this can significantly influence resulting mucocutaneous symptoms experienced by the users.

Reply

We agree that the nature of biocide/pesticide exposure can influence mucocutaneous outcomes. Accordingly, we clarified Section 2.4 to describe how pesticide exposure was characterized in this study. Exposure information was collected using an interviewer-administered questionnaire and categorized by pesticide type (e.g., insecticides, herbicides, fungicides, and molluscicides). Information on specific active ingredients or formulations was not collected and is acknowledged as a study limitation.

4.2 The statement on ‘two specialists in dermatology’ is redundantly presented, see lines 149 – 150.

Reply

Thank you for noting this redundancy. We removed the repeated statement regarding “two specialists in dermatology” (Lines 149–150) and revised the sentence for clarity to avoid duplication.

4.3 The authors should specify which language was used during the interview.

Reply

Thank you for this comment. We have revised the Materials and Methods section to specify the interview language. All interviews were conducted in Thai by trained research assistants using a standardized interviewer-administered questionnaire.

4.4 Lines 169 – 171: review and improve the sentence formulation.

Reply

Thank you for this comment. We revised the sentences to improve sentence clarity and flow by streamlining the description of interviewer training and standardization and removing redundant phrasing.

4.5 The authors calculated a minimum sample size of 228 participants. But obtained only 226 individuals. This therefore limits the statistical power of the study, and questions the conclusions of the findings.

Reply

Thank you for this comment. The calculated minimum sample size of 228 participants already included an allowance for 15% missing data, as stated in the Methods section. A total of 226 participants completed the study and were included in the final analysis, representing a shortfall of only two participants (<1%). This small difference is unlikely to materially affect the precision of prevalence estimates or the overall interpretation of findings in this descriptive cross-sectional study.

4.6 The main crop cultivated, and distance from pesticide-treated field to home should be part of the information presented in the questionnaire.

Reply

Thank you for this suggestion. We have revised Section 2.4 to specify that the questionnaire captured the main cultivated crop(s) and the approximate distance from the participant’s home to the nearest pesticide-treated field as additional exposure-context variables.

4.7 The section on sample size should be presented earlier and separated from that of Statistical analysis.

Reply

Thank you for this comment. We have revised the Methods section to present the sample size calculation earlier as a separate subsection, preceding the Statistical Analysis section.

4.8 The section Statistical analysis is poorly done with sentence without verb. There is a mixture of qualitative and quantitative data expression, and all these were grouped into descriptive analysis. Such an inconsistency should be revisited.

Reply

Thank you for this comment. We have revised the Statistical Analysis section to correct sentence structure, clarify the distinction between categorical and continuous variables, and ensure consistent and accurate description of descriptive statistical methods.

  1. Results and Discussion

5.1 Sub-titles should be added to the results section.

Reply

Thank you for this comment. Subtitles have now been added to the Results section to improve clarity and organization.

5.2 The Table 1 should be reorganized with age range, exposure duration levels, educational levels, both sexes, etc. Also, data on participants’ race, education, and employment history should be presented.

Reply

Thank you for this valuable suggestion. We have revised Table 1 as recommended by reorganizing participant characteristics and adding age range and categories, sex distribution, exposure duration levels, crop-related variables, and other occupational exposure information. Educational level and detailed employment history were not collected in this study, as the primary aim was to describe pesticide exposure and mucocutaneous outcomes rather than educational or interventional factors. However, we agree that these variables are important for understanding risk perception and protective behaviors, and we have acknowledged their absence as a limitation and highlighted their inclusion as key components in future studies, particularly those involving educational or behavioral interventions.

5.3 Line 225, the word ‘reasonable’ should corrected to ‘good, very good, etc.’

Reply

Thank you for this comment. We have revised the wording by replacing “reasonable” with “good” to more accurately describe the reported post-pesticide hygiene behaviors.

5.4 Table 2 -4: present it as a 3 columns table, with the 1st being the variable.

Reply

Thank you for this comment. Tables 2–4 have been reformatted into three-column tables, with the first column presenting the variable, the second the category, and the third the corresponding values, to improve clarity and consistency.

5.5 Table 2: the route of pesticide exposure signifies the way through which this chemical enters the human system. I suggest the authors to rather use ‘the mean of pesticide application’.

Reply

Thank you for this helpful suggestion. We have revised Table 2 by replacing “route of pesticide exposure” with “method of pesticide application” to more accurately reflect the application practices reported by participants.

5.6 Lines 230 – 241: quantitative figures should be added when commenting the findings.

Reply

Thank you for this comment. We have revised Lines 230–241 to include relevant quantitative figures (n and percentages) when describing the observed skin, ocular, and oral/nasal symptoms, in accordance with the data presented in Tables 3 and 4.

5.7 Out of the 226 farmers interviewed, only 53 applied biocides. It will be important to find out whether the remaining portion used or were directly or indirectly exposed biocides, if yes how?

Reply

Thank you for this important comment. In this study, pesticide applicators were defined as participants who reported personally mixing/loading and applying pesticides (n = 53). Indirect exposure pathways among non-applicators were not systematically assessed; we have acknowledged this as a limitation in the revised manuscript.

5.8 The authors should present the findings on intermittent mucocutaneous symptoms pruritic, erythematous  skin

Reply

Thank you for this comment. We have revised the Results section to explicitly present the quantitative findings supporting the intermittent pattern of pesticide-attributed pruritic and erythematous skin symptoms (itching 90.9%, erythema 78.8%, and symptoms occurring ‘sometimes’ in 72.7%; Table 3).

5.9 As part of the findings, the authors should present how independent variable such age, sex, education, etc. influence other factors (mucocutaneous symptoms, DLQI).

Reply

Thank you for this suggestion. We conducted additional analyses to examine associations between participant characteristics (age and sex) and mucocutaneous symptoms and DLQI, and these results have been added to the Results section. The analytical approach has been clarified in the Methods section, and the absence of educational level data has been explicitly acknowledged as a study limitation.

5.10 The mucocutaneous symptoms are susceptible to be influenced by the type of pesticide used. The authors should present findings with respect to this.

Reply

Thank you for this important comment. We performed additional analyses among participants who personally applied pesticides to assess associations between mucocutaneous symptoms, DLQI, and pesticide class (herbicides, insecticides, and fungicides). The corresponding methods, results, and interpretation have been added to the Methods, Results, and Discussion sections, respectively.

6 Conclusions

6.4 The key findings of the study should be clearly presented prior to recommendations.

Reply

Thank you for this comment. We have revised the Conclusions section to clearly present the key study findings before outlining recommendations, thereby improving clarity and logical flow.

7 General comment

7.1 Review the English language, harmonizing the caps lock and abbreviations. The findings should be presented using the past tense.

Reply

Thank you for this comment. The manuscript has been revised to improve English language quality, harmonize capitalization and abbreviation usage, and ensure that findings are consistently presented in the past tense. Professional English editing was performed (MDPI English Editing).

7.2 The section Highlights is not an integral section of the author’s guidelines of IJERPH.

Reply

Thank you for this comment. We acknowledge that the Highlights section is not mandatory in IJERPH. However, as the Editorial Office has recently introduced Highlights as an optional element and indicated that it may improve visibility and discoverability, we have retained this section in the revised manuscript in accordance with their guidance.

8 References

8.1 This section should be reviewed, harmonized according to IJERPH the author’s guidelines.

Reply

Thank you for this comment. The References section has been reviewed and harmonized to ensure consistency and completeness. In accordance with IJERPH’s current free-format submission policy, references are presented in a consistent style and include all required bibliographic elements. Reference management was performed using EndNote, and the manuscript will be fully formatted according to IJERPH guidelines at the production stage, if accepted.

Round 2

Reviewer 2 Report

Comments and Suggestions for Authors

Dear authors,

The study investigated the levels and dermatology-specific impacts of pesticide-attributed symptoms, notably mucocutaneous symptoms among Thai agricultural workers. It is relevant with important implications in public health management of pesticide toxicity among farmers. The authors are encouraged to improve the quality of the manuscript using the guidelines below in addition to few remarks made in the main document. Globally, the manuscript requires minor revisions to meet publication standards. The authors are encouraged to improve the quality of the manuscript by addressing the detailed comments below and the specific remarks provided in the main document.

-------------------

  1. Line 157: the authors indicated that they collected data on participant race. I wonder whether this should not be corrected to ethnicity.
  2. Redundancy between lines 169 – 170 and lines 173 – 174, lines 168 – 169 and 175 – 176.
  3. Lines 199 – 200, the sentence presented is not part of the sub-section sample size calculation. It should be taken to the appropriate section.
  4. Table 2: do format by merging some cells of the column ‘characteristics’ as indicate in the main document. The same formatting should be done on tables 3 & 4.

Author Response

  1. Line 157: the authors indicated that they collected data on participant race. I wonder whether this should not be corrected to ethnicity.

Reply

Thank you for this comment. We agree that ‘ethnicity’ is the more appropriate term in this setting. We have replaced ‘race’ with ‘ethnicity’ throughout and clarified that all participants identified as Thai; therefore, ethnicity was not included in the analytical models.

  1. Redundancy between lines 169 – 170 and lines 173 – 174, lines 168 – 169 and 175 – 176.

Reply

Thank you for this comment. We agree that these statements were repetitive. We have revised the manuscript by removing the duplicated descriptions of (1) the use of trained interviewers/standardized protocol and (2) the approximately 30-minute interview duration, retaining each detail only once in the Data Collection section to improve conciseness and clarity.

  1. Lines 199 – 200, the sentence presented is not part of the sub-section sample size calculation. It should be taken to the appropriate section.

Reply

Thank you for this comment. We agree that the sentence in Lines 199–200 describes planned analyses rather than sample size determination. We have removed it from the Sample Size Calculation subsection and relocated it to the Statistical Analysis subsection.

  1. Table 2: do format by merging some cells of the column ‘characteristics’ as indicate in the main document. The same formatting should be done on tables 3 & 4.

Reply

Thank you for this suggestion. We revised Tables 2–4 by merging the relevant cells in the ‘Characteristics’ column to group related items and improve readability, consistent with the formatting in the main document.